# Dual-Route Mental Imagery for Robust VLM-based Medical Image Diagnosis

## Abstract

Despite rapid progress of large vision-language models (VLMs), their diagnostic predictions in medical imaging remain brittle and often clinically inconsistent. Inspired by how radiologists rely on prototype-based mental imagery, we propose **Dual-Route Mental Imagery**, the first framework that formalizes prototype-conditioned reasoning for VLMs. Our method conditions diagnosis on (patient, prototype) pairs, instantiating two complementary reasoning routes—healthy and diseased—that yield interpretable reference-level traces and expose uncertainty when the two routes conflict. On chest X-ray benchmarks, our approach delivers substantial gains: on the Kermany dataset, it achieves 92.6% accuracy, on par with the expert-designed network LungConVT-Net, and further improves to 95.9% with uncertainty handling, while substantially outperforming single-image VLM inference. These results demonstrate that prototype-guided dual-route mental imagery not only enhances the robustness and accuracy of VLM-based diagnosis, but also provides a novel bridge between cognitive science and AI for healthcare.

## 1 Introduction

Large vision-language models (VLMs) Wang et al. (2024) have recently demonstrated impressive capabilities in multimodal reasoning and open-ended medical image interpretation. By combining natural language generation with visual understanding, they open new opportunities for VLM-assisted diagnosis, particularly in chest X-ray analysis Han et al. (2024); Huang et al. (2023), where textual reasoning can enhance transparency and clinical adoption. However, despite rapid progress, VLMs remain brittle in chest X-ray diagnosis Li et al. (2023b): their predictions are often overconfident, inconsistent with clinical reasoning, and highly sensitive to distribution shifts Ktena et al. (2024); Guan & Liu (2022); Yang et al. (2024). These limitations significantly constrain their reliability in safety-critical healthcare applications.

In contrast, radiologists rarely rely on a single image for diagnosis. Instead, they often engage in mental imagery Azizi et al. (2023); Zhang et al. (2023): retrieving and comparing the current case against prototypical exemplars of both healthy and diseased states. Such prototype-conditioned reasoning helps resolve ambiguity, calibrate decisions, and provide interpretable evidence for clinical judgment Nauta et al. (2023). Yet current VLMs lack mechanisms to emulate this human-like prototype-guided comparison process, leading to limited robustness and interpretability.

To bridge this gap, we propose Dual-Route Mental Imagery, the first framework that formalizes prototype-conditioned reasoning for VLM-based diagnosis. Specifically, we construct curated prototype libraries of chest X-rays, selected through sharpness filtering and clustering-based representativeness. By conditioning on (patient, prototype) pairs, our method instantiates two complementary reasoning routes—healthy and diseased—that yield reference-level interpretability and expose uncertainty when the two routes strongly diverge. This dual-route mechanism moves beyond direct pattern recognition in standard VLM inference, embedding a cognitively inspired recall-and-compare process that more closely mirrors human diagnostic reasoning.

We validate our framework on standard chest X-ray benchmarks. Using Bagel as a representative VLM baseline, our method achieves 92.6% accuracy on the Kermany dataset, matching the expert-designed network LungConVT-Net (92.6%), and further improves to 95.9% with uncertainty handling, substantially surpassing Bagel's single-image inference (90.1%). On the Kaggle Chest

X-ray dataset, our approach also demonstrates strong generalization, reaching 91.3% accuracy and 93.6% with uncertainty handling, approaching expert-level performance (95.3%) while greatly outperforming Bagel (80.5%).

Our contributions are summarized as follows:

1. Prototype-conditioned dual-route reasoning. We propose the first framework that formalizes radiologists' mental imagery into VLM reasoning, linking cognitive science with the design of VLM reasoning mechanisms.

2. A novel training paradigm. By conditioning training on balanced (patient, prototype) pairs from curated prototype sets, we enforce a recall-and-compare process that enables VLMs to reason by contrasting patients against representative exemplars.

3. Enhanced robustness, accuracy, and interpretability. Our method substantially outperforms Bagel as a single-image VLM baseline in chest X-ray diagnosis, produces reference-level reasoning traces, and introduces an uncertainty mechanism that improves safety in high-stakes medical decisions.

In summary, prototype-guided dual-route mental imagery significantly improves the robustness and clinical reliability of VLM-based diagnosis, providing a novel bridge between cognitive science and AI for healthcare.

## 2 RELATED WORKS

### 2.1 VISION-LANGUAGE MODELS FOR MEDICAL IMAGE ANALYSIS.

Large vision-language models such as LLaVA Li et al. (2023a), Qwen-VL Bai et al. (2023), and closed-source VLMs like GPT-4V and Gemini Qi et al. (2023); Huang et al. (2021); Ryu et al. (2025) have demonstrated strong capabilities in multimodal reasoning and free-form medical image interpretation. Their use in chest radiography has been explored for report generation Wu et al. (2024), question answering Moor et al. (2023), and diagnosis Tiu et al. (2022). Yet, despite this flexibility, VLMs remain brittle in clinical settings: predictions are often overconfident, sensitive to distribution shifts Yang et al. (2024); Fehr et al. (2024); Ktena et al. (2024), and misaligned with radiological reasoning. Expert-designed architectures such as LungConVT-Net Lasker et al. (2026) achieve strong performance but rely on supervised training with limited adaptability and interpretability. Our work diverges by enhancing VLM robustness while explicitly aligning inference with clinical diagnostic practice.

### 2.2 MENTAL IMAGERY, INTERPRETABILITY, AND UNCERTAINTY IN MEDICAL AI.

Cognitive science suggests that radiologists frequently rely on mental imagery Azizi et al. (2023); Zhang et al. (2023)—retrieving and comparing current cases with prototypical exemplars of both healthy and diseased states—to resolve ambiguity, calibrate confidence, and justify decisions. This contrasts sharply with current VLMs, which lack mechanisms to emulate such recall-and-compare processes. Prior efforts toward interpretability (e.g., saliency maps, attention visualization Abdar et al. (2021); Nehme et al. (2023)) and uncertainty estimation (e.g., Bayesian deep learning, ensembles Fehr et al. (2024); Zhu et al. (2021)) provide useful but indirect signals, often disconnected from clinicians' reasoning. Our framework, *Dual-Route Mental Imagery*, bridges this gap by formalizing prototype-conditioned mental imagery within VLM inference. By instantiating complementary "healthy" and "diseased" reasoning routes, it generates reference-level reasoning traces and naturally exposes uncertainty when the two routes diverge Ullah et al. (2025); Nehme et al. (2023)—thereby grounding robustness, interpretability, and reliability in a cognitively inspired process.

### 2.3 CHALLENGES OF DOMAIN SHIFT AND CLINICAL ADOPTION.

A longstanding challenge in medical AI is distribution shift Guan & Liu (2022); Ktena et al. (2024); Godau et al. (2025): diagnostic models often degrade when deployed across hospitals, imaging devices, or patient populations that differ from the training data. Prior work has investigated domain adaptation Guan & Liu (2022); Azizi et al. (2021), self-supervised pretraining Huang et al.

(2023); Han et al. (2024), and ensemble approaches Abdar et al. (2021) to improve generalization, but robustness in real-world clinical environments remains elusive. Clinical adoption further requires more than accuracy: physicians demand models that expose uncertainty Fehr et al. (2024); Han et al. (2024), provide interpretable reasoning, and integrate seamlessly with diagnostic workflows Bhayana (2024). In our study, we explicitly assess generalization by validating on both the Kermany and Kaggle Chest X-ray datasets, demonstrating that dual-route mental imagery substantially improves cross-dataset robustness. By grounding predictions in prototype-based comparisons and surfacing uncertainty when reasoning routes diverge, our framework not only strengthens generalization but also enhances transparency and reliability in a cognitively inspired manner.

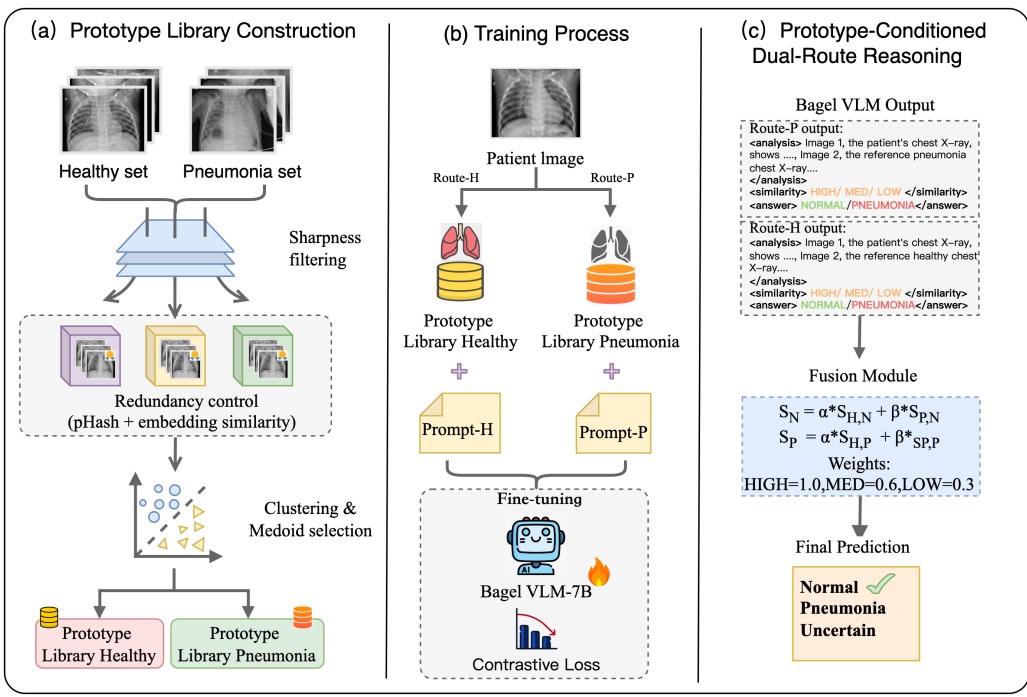

Figure 1: **Overview of the proposed *Dual-Route Mental Imagery* framework.** (a) **Prototype library construction.** From all chest X-rays, we filter blurred scans with Laplacian sharpness, remove redundant cases via perceptual hashing and embedding similarity, and apply clustering with medoid selection to obtain compact libraries of representative *healthy* and *pneumonia* prototypes. (b) **Training with prototype pairs.** The model is trained on *(patient, prototype)* pairs with structured prompts rather than single images, encouraging recall-and-compare reasoning and implicitly enforcing contrastive supervision. (c) **Dual-route reasoning.** At inference, the patient image is paired with top-$M$ prototypes from both libraries to instantiate *Healthy* and *Pneumonia* routes. Each produces analysis, similarity judgments, and predictions, which are fused with similarity-weighted aggregation. Divergence between the routes naturally signals diagnostic uncertainty.

## 3 METHOD

### 3.1 OVERVIEW

To bridge human-like reasoning and VLM-based diagnosis, we propose the *Dual-Route Mental Imagery* framework (Figure 1). It grounds predictions in structured prototype comparisons.

The pipeline begins with constructing curated prototype libraries of healthy and pneumonia exemplars through sharpness filtering, redundancy removal, and clustering (Fig. 1a).

During training, the model is conditioned on *(patient, prototype)* pairs via structured prompts, thereby shifting the paradigm from single-image learning to pairwise recall-and-compare reasoning (Fig. 1b).

At inference, two complementary routes are instantiated: the **Healthy Route**, which compares the patient image against prototypical healthy references, and the **Pneumonia Route**, which compares against pneumonia prototypes (Fig. 1c). Each route yields structured analyses, similarity assessments, and diagnostic predictions. Their outputs are fused using similarity-based weighting, while divergence between the two routes is explicitly surfaced as uncertainty.

This design explicitly mirrors the diagnostic strategy of radiologists—recalling and contrasting prototypical cases—and grounds robustness, interpretability, and clinical reliability in a cognitively inspired process.

### 3.2 PROTOTYPE LIBRARY CONSTRUCTION

To obtain compact yet representative prototype libraries for both healthy and pneumonia cases, we design a three-stage construction framework (Fig. 1a). This procedure aims to balance *diagnostic reliability*, *redundancy reduction*, and *coverage of diverse imaging patterns*.

**(1) Sharpness-aware filtering.** To prevent low-quality scans from compromising clinical interpretability, we first assess image clarity using the variance of the Laplacian:

$$s(I) = \text{Var}(\nabla^2 I),$$

where $I$ denotes a chest X-ray image and $s(I)$ its sharpness score. Images with higher $s(I)$ are preferentially retained, while globally blurred scans are discarded, ensuring that prototypes are built upon diagnostically reliable inputs.

**(2) Multi-stage redundancy control.** To avoid dominance of near-duplicates in the prototype set, we introduce a two-level pruning strategy. Candidate duplicates are first grouped via perceptual hashing (pHash, coarse similarity). Within each group, we compute $\ell_2$-normalized ResNet-50 embeddings $\mathbf{f} \in \mathbb{R}^{2048}$ and apply cosine similarity:

$$\cos(\mathbf{f}_i, \mathbf{f}_j) = \frac{\mathbf{f}_i \cdot \mathbf{f}_j}{\|\mathbf{f}_i\|_2 \|\mathbf{f}_j\|_2}.$$

Samples with $\cos(\mathbf{f}_i, \mathbf{f}_j) > \tau$ (threshold $\tau$) are considered redundant, with the sharper instance always retained. This design balances compactness and diversity.

**(3) Coverage-preserving clustering.** We then apply MiniBatchKMeans to partition the remaining cases into $k$ clusters, each representing a distinctive imaging pattern. For each cluster $C_c$, we select as prototype the exemplar closest to its normalized centroid:

$$m_c = \arg \max_{x \in C_c} \cos(\mathbf{f}_x, \hat{\boldsymbol{\mu}}_c), \quad \hat{\boldsymbol{\mu}}_c = \frac{\boldsymbol{\mu}_c}{\|\boldsymbol{\mu}_c\|_2}.$$

If a cluster is empty, we back-fill with cases least similar to existing prototypes, ensuring exactly $k$ representatives with broad coverage.

Through this three-stage process, we obtain two prototype libraries, $\mathcal{P}_H$ (healthy) and $\mathcal{P}_P$ (pneumonia). These libraries serve as cognitively inspired exemplars, supporting dual-route reasoning by anchoring decisions on both prototypical similarity and feature-based analysis.

### 3.3 TRAINING WITH PROTOTYPE-BASED PAIRS

To shift the model from direct pattern recognition toward recall-and-compare reasoning, we train the VLM on prototype pairs rather than single images (Fig. 1b). Each training instance is a structured pair $(I, p)$, where $I$ is a patient image and $p$ is a prototype drawn from the curated libraries. The pair is encoded with prompts that explicitly instruct the model to compare the two images and reason about their similarities and differences.

**(1) Balanced pairing.** For each patient image, we sample prototypes from both the healthy and pneumonia libraries to construct positive and negative pairs. Positive pairs $(I, p^+)$ align with the

patient's ground-truth label (e.g., a pneumonia case with a pneumonia prototype), while negative pairs $(I, p^-)$ come from the opposite library. This balanced design exposes the model to both concordant and discordant exemplars, ensuring that learning is conditioned on explicit contrasts.

**(2) Implicit contrastive supervision.** The model is optimized with a standard cross-entropy loss on categorical predictions (NORMAL, PNEUMONIA). Although no additional loss terms are introduced, the balanced positive/negative sampling implicitly enforces a contrastive signal: predictions are rewarded when aligned with matched prototypes and penalized when inconsistent with mismatched ones. This mechanism grounds the VLM's reasoning in prototype-based comparisons rather than isolated visual patterns.

**(3) Structured prompting effect.** Because each pair is prompted to generate not only a categorical diagnosis but also a similarity judgment and textual analysis, the supervision on the final label indirectly regularizes these intermediate outputs. The model learns to treat prototypes as anchors for reasoning, linking interpretable reference-level comparisons with the ultimate diagnostic decision.

In summary, training with prototype pairs instills a recall-and-compare paradigm in the VLM. This design enforces prototype-conditioned reasoning during training and prepares the model for robust dual-route inference.

### 3.4 PROTOTYPE-GUIDED DUAL-ROUTE REASONING

To emulate radiologists' recall-and-compare diagnostic strategy, we design a dual-route prototype-conditioned reasoning mechanism (Fig. 1c), where each route retrieves reference prototypes, conditions the VLM with structured prompts, and aggregates predictions into a fused decision.

**(1) Prototype retrieval.** For each patient image $I$, we retrieve the top-$M$ prototypes from both libraries, denoted as $\{p_H^1, \ldots, p_H^M\} \subset \mathcal{P}_H$ and $\{p_P^1, \ldots, p_P^M\} \subset \mathcal{P}_P$, where $M$ controls the number of references. Selection is based on a joint score combining embedding similarity and prototype sharpness:

$$\text{score}(I, p) = \lambda \cdot \cos\big(f(I), f(p)\big) + (1 - \lambda) \cdot z\big(s(p)\big),$$

where $f(\cdot)$ denotes ResNet-50 embeddings, $s(\cdot)$ the Laplacian variance (sharpness), and $z(\cdot)$ the z-scoring function. The parameter $\lambda \in [0, 1]$ balances semantic similarity and image clarity.

**(2) Route-specific reasoning.** Each prototype is paired with the patient image and fed into the Bagel VLM using structured prompts. Two complementary routes are instantiated:

- **Route-H:** patient vs healthy prototype,
- **Route-P:** patient vs pneumonia prototype.

For each pair, the model outputs: (i) a textual analysis (<analysis>), (ii) a similarity judgment (<similarity_to_reference> $\in$ {HIGH, MEDIUM, LOW}), and (iii) a categorical diagnosis (<answer> $\in$ {NORMAL, PNEUMONIA}).

These structured prompts enforce contrastive reasoning, ensuring that both textual analysis and similarity judgments contribute directly to downstream scoring.

**(3) Route-level aggregation.** Similarity levels are mapped to numerical weights, with higher weights assigned to stronger matches. For Route-H, we compute weighted scores:

$$S_{H,N} = \sum_{m=1}^{M} w_m \cdot \mathbf{1}[\hat{y}_m = \text{NORMAL}], \quad S_{H,P} = \sum_{m=1}^{M} w_m \cdot \mathbf{1}[\hat{y}_m = \text{PNEUMONIA}],$$

with $(S_{P,N}, S_{P,P})$ defined analogously for Route-P.

**(4) Dual-route fusion and uncertainty.** The final decision is obtained via:

$$S_N = \alpha S_{H,N} + \beta S_{P,N}, \quad S_P = \alpha S_{H,P} + \beta S_{P,P},$$

where $\alpha, \beta > 0$ control the relative contributions of the two routes. The default prediction is given by comparing $S_N$ and $S_P$:

$$\hat{y} = \begin{cases} \text{PNEUMONIA}, & S_P > S_N, \\ \text{NORMAL}, & \text{otherwise}, \end{cases}$$

which corresponds to a deterministic tie-breaking rule (ties default to NORMAL).

Importantly, this operator is configurable: replacing the inequality with $S_P \geq S_N$ or $S_N \geq S_P$ systematically biases the model toward higher sensitivity (favoring PNEUMONIA) or higher specificity (favoring NORMAL), without introducing additional heuristics. Moreover, ties ($S_P = S_N$) or contradictory predictions between Route-H and Route-P can be flagged as **uncertain**.

This flexibility enables the framework to adapt decision policies to clinical requirements, systematically trading off sensitivity, specificity, and uncertainty handling.

# 4 EXPERIMENTS

## 4.1 EXPERIMENTAL SETUP

**Datasets.** We evaluate our framework on the Kermany chest X-ray dataset released by Kermany et al. Kermany (2018). The dataset contains chest radiographs from patients aged 1–5, categorized into NORMAL and PNEUMONIA. Following standard practice, we merge the official training and validation sets (3883 pneumonia, 1349 normal) for training, and retain the original test split (390 pneumonia, 234 normal) for evaluation.

To construct prototype libraries, we apply the procedure in Sec. 3.2. Specifically, we select 256 NORMAL and 512 PNEUMONIA prototypes, which are then paired with patient images at training time. At inference, we evaluate dual-route reasoning with varying numbers of retrieved prototypes ($M = 1, 2, 3$).

## 4.2 IMPLEMENTATION DETAILS

**Prototype Library Construction.** All chest X-rays are preprocessed into $384 \times 384$ resolution and embedded with ResNet-50 features. We apply sharpness-aware filtering, two-stage redundancy pruning (pHash + cosine similarity with $\tau = 0.985$), and MiniBatchKMeans clustering to obtain compact yet diverse prototype sets. The final libraries contain 256 NORMAL and 512 PNEUMONIA exemplars.

**Training with Prototype-Based Pairs.** We finetune **Bagel-7B-MoT** using `torchrun` on 2 A100 GPUs. Training runs for 750 steps with 75 warmup steps. The effective batch size is $\sim$16k tokens per step (`expected_num_tokens=16384`), optimized with AdamW at a learning rate of $1 \times 10^{-5}$ under a cosine scheduler. CPU offloading is enabled to improve memory efficiency.

**Dual-Route Prototype-Conditioned Reasoning.** At inference, each test image retrieves the top-$M$ prototypes ($M = 1, 2, 3$) per route based on a joint score of embedding similarity and sharpness ($\lambda = 0.85$). Route-level predictions are fused via weighted aggregation with $\alpha = \beta = 1.0$.

All experiments are implemented in Python with fixed random seeds, and no further hyperparameter tuning is performed beyond the above settings.

## 4.3 EVALUATION METRICS

We evaluate our framework and baselines with standard diagnostic and selective prediction metrics.

**Accuracy (Acc)**: overall proportion of correctly classified cases.

**Sensitivity (Sens, Recall)**: true positive rate for pneumonia, capturing the ability to detect pathological cases.

**Specificity (Spec)**: true negative rate for normal cases, reflecting robustness against false alarms.

**Balanced Accuracy (BA)**: average of sensitivity and specificity, accounting for class imbalance.

**Macro-F1**: unweighted mean of per-class F1 scores, balancing precision and recall.

For our dual-route framework, we further examine **tie-breaking policies** (tie$\rightarrow$N vs. tie$\rightarrow$P), which determine the predicted label when the two reasoning routes disagree. These policies enable explicit control of the sensitivity–specificity trade-off.

Under the selective prediction setting, we report **Coverage** (the fraction of test cases with confident predictions) and **Acc@C / Sens@C / Spec@C / Macro-F1@C**, which are computed only on the retained subset. This reflects the practical value of abstaining on uncertain cases: higher reliability on auto-reported predictions while deferring ambiguous cases to human experts.

## 4.4 MAIN RESULTS

| Model | Acc | Sens | Spec | BA | Macro-F1 |
|---|---|---|---|---|---|
| Qwen2.5-VL-7B | 0.672 | 0.931 | 0.239 | 0.585 | 0.567 |
| LLaVA-Next-13B | 0.704 | 0.828 | 0.496 | 0.662 | 0.667 |
| Bagel (single) | 0.901 | 0.864 | **0.962** | 0.913 | 0.898 |
| LungConVT-Net | **0.926** | **0.982** | 0.833 | 0.908 | 0.919 |
| Ours (M=2, tie→P) | **0.926** | 0.962 | 0.868 | **0.915** | **0.942** |

Table 1: Overall performance comparison on the Kermany pediatric chest X-ray dataset. Best values in each column are highlighted in **bold**. Our method achieves the most balanced performance across metrics.

**Overall comparison.** Table 1 compares our framework with representative VLMs (Qwen2.5-VL-7B, LLaVA-Next-13B, Bagel) in the single-image setting, as well as the expert-designed CNN–Transformer hybrid LungConVT-Net.

The results reveal several key findings. First, general-purpose VLMs (Qwen2.5-VL-7B and LLaVA-Next-13B) struggle in this setting: although Qwen2.5-VL-7B achieves relatively high sensitivity (0.931), its specificity collapses to only 0.239, highlighting severe over-diagnosis of pneumonia. LLaVA-Next-13B improves specificity (0.496) but sacrifices sensitivity (0.828), confirming that single-image VLM inference cannot balance pathological detection and false positive control. Second, Bagel, a stronger medical VLM, attains 0.901 accuracy and a state-of-the-art specificity of 0.962, but its sensitivity (0.864) is markedly lower than both Qwen2.5-VL-7B and LungConVT-Net, leading to missed pneumonia cases. Third, LungConVT-Net achieves excellent sensitivity (0.982), consistent with its design focus on pathology detection, but at the cost of reduced specificity (0.833).

By contrast, our dual-route framework (M=2, tie→P) attains the same top accuracy as LungConVT-Net (0.926), but substantially improves specificity (0.868 vs. 0.833) while maintaining strong sensitivity (0.962). Most importantly, it achieves the highest Balanced Accuracy (0.915) and Macro-F1 (0.942) among all models. This indicates that our method delivers the most balanced diagnostic capability, avoiding the pitfalls of both false alarms (low specificity) and missed cases (low sensitivity). These improvements highlight the clinical value of grounding decisions in prototype-based dual-route reasoning, further reflecting the advantages of simulating expert diagnostic strategies.

**Ablation on prototype retrieval.** We further investigate the impact of prototype retrieval count ($M$) and tie-breaking policies, as shown in Table 2.

The results show a clear trade-off between sensitivity and specificity. When ties are resolved toward NORMAL, specificity peaks at 0.983 (M=1), but sensitivity drops sharply to 0.805. Conversely, tie→PNEUMONIA consistently boosts sensitivity to 0.962, but lowers specificity (0.863–0.868). This controllable behavior demonstrates that prototype-conditioned reasoning allows our method to be tailored: prioritizing high sensitivity for safer pneumonia screening, or high specificity to minimize false positives in low-prevalence settings.

Importantly, the best overall performance emerges with M=2 or M=3 under tie→P, both yielding 0.926 accuracy, 0.962 sensitivity, 0.868 specificity, and the highest Macro-F1 (0.942). This configuration achieves a clinically desirable balance, ensuring pneumonia is rarely missed while false positives are controlled. The stability of results across M=2–4 also suggests that our dual-route design is robust to the choice of retrieval depth: adding more prototypes does not destabilize predictions, unlike conventional k-nearest neighbor–style retrieval which may be sensitive to noise. This robustness is an appealing property for real-world deployment, where retrieval hyperparameters must remain simple and reliable.

| $M$ | Tie Policy | Acc | Sens | Spec | Macro-F1 |
|---|---|---|---|---|---|
| 1 | tie→N | 0.872 | 0.805 | **0.983** | 0.887 |
| 1 | tie→P | 0.925 | **0.962** | 0.863 | 0.941 |
| 2 | tie→N | 0.885 | 0.833 | 0.970 | 0.900 |
| 2 | tie→P | **0.926** | **0.962** | 0.868 | **0.942** |
| 3 | tie→N | 0.897 | 0.856 | 0.966 | 0.913 |
| 3 | tie→P | **0.926** | **0.962** | 0.868 | **0.942** |
| 4 | tie→N | 0.897 | 0.856 | 0.966 | 0.913 |
| 4 | tie→P | **0.926** | **0.962** | 0.868 | **0.942** |

Table 2: Ablation study on the number of retrieved prototypes $M$ and tie-breaking policies. Best values in each column are highlighted in **bold**.

| $M$ | Coverage | Acc@C | Sens@C | Spec@C | Macro-F1@C |
|---|---|---|---|---|---|
| 1 | 85.7% | **0.964** | 0.954 | **0.981** | **0.971** |
| 2 | 88.1% | 0.960 | 0.956 | 0.967 | 0.967 |
| 3 | **89.7%** | 0.959 | **0.957** | 0.962 | 0.967 |

Table 3: Selective prediction results on the Kermany chest X-ray dataset. Coverage denotes the proportion of samples with confident predictions; metrics with @C are computed only on this subset. Best values in each column are highlighted in **bold**.

**Selective prediction.** We assess the selective prediction setting, where the method abstains from uncertain cases flagged by divergent dual-route predictions; the results are summarized in Table 3.

This evaluation simulates clinical deployment: ambiguous cases can be deferred to radiologists, while high-confidence predictions are auto-reported. The results in Table 3 show that even with abstention, coverage remains high: 85.7% to 89.7% of cases are classified automatically. Crucially, performance on this confident subset is substantially boosted. For instance, with $M = 1$, accuracy on certain cases reaches 0.964 and Macro-F1 climbs to 0.971, a significant improvement over overall performance without abstention. As M increases, coverage improves monotonically (85.7% → 89.7%), while Acc@C, Sens@C, and Spec@C stabilize near 0.96–0.97, indicating both reliability and consistency.

This result highlights a unique advantage of our framework: the abstention mechanism emerges naturally from dual-route divergence, requiring no auxiliary uncertainty estimation module. It provides a principled way to balance automation and safety—most cases can be diagnosed with near-expert reliability, while ambiguous cases are transparently flagged for human review. Such selective prediction capabilities are particularly valuable for clinical adoption, where trustworthiness and interpretability are as critical as accuracy.

### 4.5 GENERALIZATION TO EXTERNAL DATASET

To assess robustness under distribution shift, we further evaluate on the Kaggle Chest X-ray dataset Chowdhury et al. (2020); Rahman et al. (2021), where 200 NORMAL and 200 PNEUMONIA images are randomly sampled. This dataset differs substantially in acquisition settings, patient demographics, and clinical context, posing a challenging out-of-domain evaluation.

The results in Table 4 reveal sharp degradation for general-purpose VLMs: Qwen2.5-VL-7B retains very high sensitivity (0.975) but its specificity collapses to only 0.360, leading to rampant over-diagnosis of pneumonia. LLaVA-Next-13B almost fails entirely, with accuracy dropping to 0.525 and specificity to just 0.050, indicating near-random predictions on normal cases. Bagel, despite being a stronger VLM, achieves only 0.805 accuracy and exhibits skewed class-wise recall (0.620 sensitivity vs. 0.990 specificity), showing that it tends to under-diagnose pneumonia. By contrast, the expert-designed LungConVT-Net remains robust, reaching 0.953 accuracy and 0.952 Macro-F1, which we use as a reference for out-of-domain evaluation.

| Model | Acc | Sens | Spec | Macro-F1 | Acc@C | Macro-F1@C |
|---|---|---|---|---|---|---|
| Qwen2.5-VL-7B | 0.668 | 0.975 | 0.360 | 0.633 | – | – |
| LLaVA-Next-13B | 0.525 | **1.000** | 0.050 | 0.387 | – | – |
| Bagel (single) | 0.805 | 0.620 | 0.990 | 0.798 | – | – |
| LungConVT-Net | **0.953** | 0.905 | **1.000** | **0.952** | – | – |
| Ours ($M = 1$, tie→N) | 0.788 | 0.590 | **0.985** | 0.735 | 0.931 | 0.915 |
| Ours ($M = 1$, tie→P) | 0.898 | **0.905** | 0.890 | 0.898 | – | – |
| Ours ($M = 2$, tie→N) | 0.813 | 0.640 | **0.985** | 0.773 | 0.933 | 0.921 |
| Ours ($M = 2$, tie→P) | 0.900 | **0.905** | 0.895 | 0.900 | – | – |
| Ours ($M = 3$, tie→N) | 0.868 | 0.755 | 0.980 | 0.851 | **0.936** | 0.929 |
| Ours ($M = 3$, tie→P) | **0.913** | **0.905** | 0.920 | **0.912** | – | – |
| Ours ($M = 4$, tie→N) | 0.870 | 0.760 | 0.980 | 0.854 | **0.936** | **0.930** |
| Ours ($M = 4$, tie→P) | **0.913** | **0.905** | 0.920 | **0.912** | – | – |

Table 4: Cross-dataset generalization results on the Kaggle Chest X-ray dataset. Best values in each column are highlighted in **bold**.

Our method demonstrates consistent advantages in this setting. Across M=1–4 under tie→N, accuracy improves steadily from 0.788 to 0.870 as retrieval depth increases, while specificity remains very high ($\geq$0.98). Under tie→P, performance peaks at 0.913 accuracy with both sensitivity and specificity balanced at $\approx$0.90 for M=3–4, sharply contrasting with the skewed trade-offs of baseline VLMs. Moreover, in the selective prediction setting, accuracy on the confident subset rises to 0.936 and Macro-F1 exceeds 0.92, while coverage stays above 93%. This indicates that our framework not only maintains reliability across all retrieval depths but also delivers near-expert performance when uncertainty is explicitly surfaced.

Crucially, our method does not force overconfident predictions on ambiguous cases: uncertainty naturally emerges when the healthy and pneumonia routes diverge, aligning with human-in-the-loop practice where straightforward cases are automated and difficult ones are deferred to radiologists. This mirrors real diagnostic workflows, enhancing both safety and trust. Another notable strength is that these gains are achieved *without any retraining or domain adaptation*. Unlike conventional single-image VLM inference, prototype-guided dual-route reasoning provides a cognitively inspired anchor that adapts more gracefully to distribution shifts. In summary, prototype-conditioned dual-route reasoning is not just a technical enhancement but a principled mechanism that unifies robustness, interpretability, and clinical alignment, paving the way for VLMs that remain reliable across heterogeneous deployment environments.

## 5 CONCLUSION

We introduced *Dual-Route Mental Imagery*, the first framework that formalizes radiologists' prototype-based mental imagery into VLM reasoning. By conditioning diagnosis on structured (patient, prototype) pairs and instantiating dual healthy–pneumonia routes, our method enforces recall-and-compare reasoning, produces interpretable reference-level traces, and transparently surfaces uncertainty when the two routes diverge. On Kermany pediatric chest X-ray dataset, it achieves balanced accuracy superior to both single-image VLMs and expert-designed networks, while selective prediction further improves reliability by abstaining on ambiguous cases. More importantly, cross-dataset evaluations on the Kaggle chest X-ray cohort demonstrate strong robustness under distribution shift without retraining, approaching expert-level performance. These findings not only advance the robustness and trustworthiness of VLM-based medical diagnosis, but also demonstrate the value of prototype-conditioned, cognitively inspired reasoning in yielding clinically aligned predictions that mirror radiologists' diagnostic strategies.

## ETHICS STATEMENT

This work leverages publicly available chest X-ray datasets (Kermany et al., Kaggle) that have been widely used in prior research. All experiments were conducted on de-identified images, and no

personally identifiable information was used. Our method is intended for research purposes only and is not validated for direct clinical deployment. We emphasize that any medical use would require rigorous regulatory approval and expert oversight.

## REPRODUCIBILITY STATEMENT

We have described datasets, prototype construction, training configurations, and evaluation protocols in detail throughout the Method and Experiments sections. Hyperparameters, preprocessing steps, and model variants are explicitly reported. We will release the related code to reproduce our experimental results, ensuring that our findings can be replicated and extended by the community.

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

# A APPENDIX

## LLM USAGE

Large Language Models (LLMs) were used to aid in the writing and polishing of the manuscript. Specifically, we used an LLM to assist in refining the language, improving readability, and ensuring clarity in various sections of the paper. The model helped with tasks such as sentence rephrasing, grammar checking, and enhancing the overall flow of the text.

It is important to note that the LLM was not involved in the ideation, research methodology, or experimental design. All research concepts, ideas, and analyses were developed and conducted by the authors. The contributions of the LLM were solely focused on improving the linguistic quality of the paper, with no involvement in the scientific content or data analysis.

The authors take full responsibility for the content of the manuscript, including any text generated or polished by the LLM. We have ensured that the LLM-generated text adheres to ethical guidelines and does not contribute to plagiarism or scientific misconduct.

