# OpenReview forum: "Dual-Route Mental Imagery for Robust VLM-based Medical Image Diagnosis"
_ICLR.cc/2026/Conference — ICLR 2026 Conference Withdrawn Submission_

### Official Review · Reviewer_9B3u · 2025-10-20

**Soundness:** 1
**Presentation:** 2
**Contribution:** 1
**Rating:** 2
**Confidence:** 4

**Summary:**

The paper proposes a method for prototype-guided VLM X-ray diagnosis. They first form a database of healthy and pneumonia patient’s chest X-rays and cluster those.
During training they pair images with prototypes of the same and opposite class, and perform cross-entropy based learning of the class prediction.
During inference the model is prompted to compare the image with prototypes of the same and opposite class, rank how similar they are and predict the class. To reach a final prediction the per pair predictions are aggregated and weighted by the similarity of the pair.
They compare their method against a few baselines and evaluate effects of different hyperparameters.

**Strengths:**

- The paper introduces a new method for prototype supported VLM diagnosis with decent clinical motivation and grounding.
- The paper is well structured.

**Weaknesses:**

1. ##### **Missing ablation of the effect of prototype inclusion**

   The core claim that the prototype-supported reasoning improves performance is not experimentally validated. The proposed method finetunes the Bagel model for the diagnosis of Pneumonia in addition to all the prototyping pipeline. What is the performance of Bagel if you simply finetune it for Pneumonia classification without including prototypes? What is the performance if you also add ensembling of multiple generations as is done in the proposed method. Currently, the paper only compares with baselines that are not finetuned on that dataset and do not use ensembling.
2. **Unsupported claims of contrastive signal**
   The paper claims that through their cross entropy training of the class label, the model also learns a contrastive signal. This is not clearly the case and I am not convinced. The model might as well ignore the prototype image and just learn to diagnose Pneumonia based on the patient image. There is no evaluation if the model becomes better at identifying similar images
3. **Unintuitive method**
   I am not sure if this method would improve the trustworthiness. For example it is not clear and rather counter-intuitive to me why a prediction of an image as “healthy” should be weighted stronger the more similar it is to an image of a pneumonia patient as would be the case following the formulas in 3.4 (3) and (4).

**Questions:**

It is unclear to me how the model is actually trained. Is the loss calculated on the final prediction of the system? If yes, how are gradients routed to the language model? Or is the loss calculated on the VLM output? If yes, how is the reasoning and similarity prediction handled? Are they precomputed, masked, or also somehow supervised?

The work is built on Bagel but it is never cited.

---

### Official Review · Reviewer_dy2H · 2025-10-31

**Soundness:** 3
**Presentation:** 3
**Contribution:** 3
**Rating:** 4
**Confidence:** 4

**Summary:**

The paper proposes the Dual-Route Mental Imagery (DRMI) framework, which integrates prototype-conditioned reasoning for medical image diagnosis using vision-language models (VLMs). By conditioning the diagnosis process on pairs of patient images and prototypical exemplars (healthy and diseased), the method emulates a cognitive process similar to how radiologists compare new cases against prototype images. This approach not only improves diagnostic robustness but also provides interpretability and exposes uncertainty in cases where the two reasoning routes (healthy and diseased) diverge. The method is evaluated on chest X-ray datasets, demonstrating strong performance and improvement over standard VLM-based methods, with enhanced accuracy, specificity, and sensitivity, especially when uncertainty handling is integrated.

**Strengths:**

- Innovative framework: Introduces a novel prototype-conditioned dual-route reasoning approach, drawing inspiration from cognitive science and radiologists' decision-making processes.

- Improved interpretability and robustness: Provides transparent reasoning and uncertainty estimation, addressing common challenges in VLM-based medical image analysis.

- Strong experimental results: Achieves competitive results on chest X-ray benchmarks (Kermany and Kaggle datasets), outperforming standard VLMs and even expert-designed networks in certain metrics.

**Weaknesses:**

- Incremental innovation: The idea of prototype-conditioned reasoning and dual-route inference is conceptually interesting, but it closely follows the pattern of previous research in medical imaging, making the contribution seem more incremental than groundbreaking.

- Limited scalability discussion: There is little analysis on how the proposed framework scales to more complex or larger datasets, or how it would handle more diverse medical imaging tasks.

- Lack of comprehensive ablation study: While the paper evaluates different prototype retrieval settings, the contributions of individual components (e.g., uncertainty handling, prototype retrieval depth) are not fully dissected, making it difficult to assess the impact of each.

**Questions:**

- How would this method perform on other medical imaging tasks, such as tumor detection or organ segmentation, where more detailed anatomical features are critical?

- Can the dual-route framework be adapted to work with more complex VLMs or multi-modal medical data that includes text reports or lab results in addition to images?

- What would the performance be on datasets with more diverse or noisy data, particularly in the presence of significant domain shifts or out-of-domain test data?

---

### Official Review · Reviewer_rTww · 2025-11-03

**Soundness:** 3
**Presentation:** 2
**Contribution:** 2
**Rating:** 4
**Confidence:** 3

**Summary:**

This paper proposes Dual-Route Mental Imagery (DRMI), a framework that integrates prototype-conditioned reasoning into Vision-Language Models (VLMs) for medical image diagnosis.  Inspired by how radiologists use mental imagery to compare current cases with prototypical healthy and diseased examples, the authors design a dual-route process:
- The Healthy Route compares the input image with healthy prototypes.
- The Diseased Route compares it with disease prototypes.

The two routes jointly produce structured reasoning traces and uncertainty signals based on their agreement or divergence.
Experiments on the Kermany and Kaggle Chest X-ray datasets demonstrate that DRMI can improve interpretability and uncertainty awareness without architectural changes.

Overall, the paper presents a conceptually interesting idea—mimicking human diagnostic reasoning via dual prototype-conditioned routes—but the methodological novelty (the underlying mechanisms—prototype retrieval, contrastive pairing, and score fusion, closely resemble established retrieval-augmented or prototype-based learning methods) and scientific rigor (absence of standard deviations, statistical tests) may fall short of ICLR’s main-track expectations.

**Strengths:**

**1. Conceptual originality and cognitive grounding**
The dual-route design (Healthy Route vs. Diseased Route) is intuitive and cognitively interpretable, drawing inspiration from human mental imagery in clinical reasoning.

**2. Method simplicity**
The method enhances reasoning ability without modifying the VLM architecture or introducing complex losses.
Performance improvement is achieved purely through structured input (patient–prototype pairs) and dual-route aggregation.

**3. Relevance to clinical deployment**
The paper explicitly emphasizes interpretability, uncertainty control, and clinical safety, making it relevant to real-world diagnostic applications.

**Weaknesses:**

**Weaknesses**

**1. Limited methodological novelty**
While the notion of “mental imagery” is appealing, the underlying mechanism closely resembles existing prototype learning or retrieval-augmented reasoning paradigms. The main innovation seems lie in task framing and input formulation.

**2. Lack of statistical significance and uncertainty validation**
The reported metrics (accuracy, F1) are presented as single averages without variance or confidence intervals.
No statistical tests (e.g., paired t-test, bootstrap CI) are provided, making it difficult to assess the reliability of the improvements.

**3. Computational cost and efficiency not discussed**
The dual-route reasoning requires retrieving top-M prototypes  for each test sample and running the VLM 2M times,  increasing inference cost.
It would strengthen the paper to include analysis of how prototype pool size affects accuracy and computational efficiency.

**4. Scalability and generalization beyond binary classification**
The framework is only evaluated on binary pneumonia vs. normal classification.
It remains unclear whether DRMI can scale to multi-label or multi-disease settings, or how prototype diversity impacts computational and memory requirements.

---

**Minor Issues**

1. Terminology inconsistency — “Dual-Route Mental Imagery” and “Prototype-Conditioned Reasoning” are used interchangeably and should be unified or clarified.
2. Fusion weights α=β=1.0 appear fixed; the paper does not evaluate alternative combinations.
3. In Figure 1(c), the arrow pointing to “Final Prediction” is unclear and should be revised.
4. In Table results, the Acc@C column under the tie→P configuration is marked “–”; clarification is needed on whether this metric was omitted or uncomputed.

**Questions:**

**1. Methodological novelty and conceptual clarity**
The paper’s central idea is compelling, but the algorithmic innovations beyond task framing remain unclear.

> **Questions:**
> - Beyond task framing, what is the core algorithmic novelty of Dual-Route Mental Imagery (DRMI)?   In what precise mathematical or operational terms does DRMI differ from prototype learning, retrieval-augmented reasoning, or case-based VLM frameworks?

---

**2. Experimental design and statistical rigor**
Reported gains  are provided without measures of dispersion, limiting conclusions about reliability.

> **Questions:**
> - Were all experiments repeated with multiple random seeds?
> - Please report mean ± standard deviation or confidence intervals, and indicate whether the improvements are statistically significant.
> - How sensitive are results to prototype selection and retrieval randomness?

---

**3. Computational and practical efficiency**
Dual-route reasoning scales inference cost linearly with the number of prototypes, requiring 2M VLM invocations per image.

> **Questions:**
> - What is the runtime and memory overhead relative to single-image Bagel inference under the same hardware and batch size?  How does latency change as M increases, and what is the accuracy–latency trade-off?
---

**4. Scalability and generalization**
Evaluation is limited to binary pneumonia vs. normal classification, leaving questions about broader applicability.

> **Questions:**
> - How would the dual-route mechanism extend to multi-class or multi-label settings?   Does performance degrade as the number/diversity of disease prototypes grows?

---

### Note · Authors · 2026-01-17

I have read and agree with the venue's withdrawal policy on behalf of myself and my co-authors.